# Single-Cell RNA Sequencing: Opportunities and Challenges for Studies on Corneal Biology in Health and Disease

**DOI:** 10.3390/cells12131808

**Published:** 2023-07-07

**Authors:** Julian A. Arts, Camille Laberthonnière, Dulce Lima Cunha, Huiqing Zhou

**Affiliations:** 1Molecular Developmental Biology, Faculty of Science, Radboud Institute for Molecular Life Sciences, Radboud University, 6525 GA Nijmegen, The Netherlands; julian.arts@ru.nl (J.A.A.);; 2Department of Human Genetics, Radboud University Medical Center, 6500 HB Nijmegen, The Netherlands

**Keywords:** RNA sequencing, single-cell, cornea, limbal stem cells

## Abstract

The structure and major cell types of the multi-layer human cornea have been extensively studied. However, various cell states in specific cell types and key genes that define the cell states are not fully understood, hindering our comprehension of corneal homeostasis, related diseases, and therapeutic discovery. Single-cell RNA sequencing is a revolutionary and powerful tool for identifying cell states within tissues such as the cornea. This review provides an overview of current single-cell RNA sequencing studies on the human cornea, highlighting similarities and differences between them, and summarizing the key genes that define corneal cell states reported in these studies. In addition, this review discusses the opportunities and challenges of using single-cell RNA sequencing to study corneal biology in health and disease.

## 1. Introduction

The cornea is a fascinating tissue located in front of the eye. It acts as a barrier to protect the eye from the environment while letting light through to enable vision [1] as a result of its unique composition of multiple transparent tissue layers (Figure 1). Together with the lens, the cornea focusses light onto the retina, functioning as a refractive system [2]. The innermost layer is a thin endothelium, important for liquid homeostasis. The endothelial cells produce a large amount of collagen fibers that form Descemet’s membrane, a thin and strong membrane acting as protection against injuries and infections, sitting on the endothelium. The thickest layer of the cornea is the stroma, which is in front of Descemet’s membrane. The stroma is composed of cells, mainly keratocytes [3], and the extracellular matrix that is produced by these keratocytes. An acellular layer, named Bowman’s layer, composed of collagen fibrils without any specific organization, is located in front of the stroma. The exact function of this layer is still under debate, given that its absence in the central part of the cornea in millions of patients who undergo laser vision correction does not affect vision [4]. The outermost layer of the cornea is the corneal epithelium, which has been extensively investigated [5,6,7]. This layer is thinner in the central region and becomes thicker towards the periphery. Surrounding the central cornea is the limbus, a circular ring, which separates the cornea from another epithelial tissue called the conjunctiva. Although the origin of the stem cells maintaining the corneal epithelium has been under debate, either surrounding the corneal epithelium or in the basal layer of the corneal epithelium in proximity to the stroma [6,8], it is generally accepted that limbal stem cells (LSCs) in the limbus represent the reservoir of stem cells that are able to regenerate the corneal epithelium upon desquamation or injury [9,10,11]. The working hypothesis is that LSCs differentiate into transient amplifying cells (TACs) that divide, and migrate towards the central cornea on the basal layer, to ultimately form the central corneal epithelium [10]. As a stratified squamous epithelium, the corneal epithelium is composed of a single layer of basal cells, followed by four or five layers of non-keratinized stratified epithelial cells. Tight-junctions and desmosomes link these cells together, making a strong physical barrier at the surface of the eye [12,13].

During embryonic development, the cornea undergoes a series of intricate molecular and cellular events to form its distinct structure. Multiple signaling pathways are involved in proper corneal morphogenesis, including the Bmp4, Wnt/β-catenin, Notch, retinoic acid, and TGF-β signaling pathways [14,15]. The outermost epithelial layers of the cornea, the central corneal epithelium, and conjunctiva, originate from the surface ectoderm [16,17,18,19]. In contrast, the stroma and endothelium have a different developmental origin; they arise from neural crest cells [16,20,21,22], and TGF-β and canonical Wnt signaling pathways are main players during their development [15,23].

While the structure of the cornea and the limbus, as well as their major cell types, have been widely investigated, the various cell states within specific cell types and their molecular signatures, especially the key marker genes that define each state, remain poorly described. This hampers our understanding of corneal homeostasis, function, and related pathology as well as exploration of therapeutic options and drug discovery.

The precise cell states can be studied by RNA sequencing, a technology that measures gene expression levels in a genome-wide scale. Recent advances in single-cell RNA sequencing [24] have enabled us to obtain these molecular insights at the single-cell level, which makes it possible to detect diverse cell states within complex tissues and organs.

The recent use of single-cell RNA sequencing has made significant contributions in eye research. Pioneering studies employing this technique have unraveled the complexity of retinal ganglion cells (RGCs) [25,26,27], which are the primary neurons responsible for transmitting visual information from the eye to the brain. Through the dissection of the transcriptomic landscape of these cells at the single-cell level, these studies uncovered the diverse cell states of RGCs, their developmental processes, and the molecular mechanisms underlying their functional specialization. These findings demonstrate the immense potential of this technique, not only in the study of RGCs but also in exploring the cornea, holding great promise for corneal research.

In this review, we provide an extensive overview of the currently published single-cell RNA sequencing studies that investigate the human cornea [28,29,30,31,32,33,34,35] and discuss the opportunities and challenges of this technology in cornea research. We chose to focus on single-cell studies of the human cornea, rather than from other model organisms [36,37,38], because human cornea data are directly relevant for regenerative medicine. In addition, how cell types in the cornea are conserved between different organisms is not yet fully clear [39,40,41].

## 2. Collection of Single-Cell Corneal RNA Sequencing Studies

So far, eight single-cell studies [28,29,30,31,32,33,34,35] containing single-cell RNA sequencing data of the human cornea have been reported (Table 1). These studies differed in their tissue collection methods, either from the whole cornea tissue, or from specific regions within the cornea, or even from larger tissues such as the whole eye. Studies by Collin et al. [30], Català et al. [31], and Ligocki et al. [32] used the complete cornea, while three other studies focused on smaller sections. Maiti et al. [33] and Dou et al. [34] investigated the central cornea, and Li et al. [35] focused on the limbus. Van Zyl et al. [28] generated data from the entire ocular anterior segment that included the cornea, iris, ciliary body, and lens. Gautam et al. [29], on the other hand, generated data from the whole eye, where a substantial amount of corneal data is available. All eight studies used the 10× Genomics platform for sequencing [42]. Among these eight studies, only van Zyl et al. [28] used single-nucleus RNA sequencing, while the other seven studies used single-cell RNA sequencing that analyzes processed RNA extracted from the cytoplasm. In this review, we refer to all studies as single-cell RNA sequencing studies, except when specifically referring to single-nucleus RNA sequencing to distinguish the method from those that analyze cytoplasmic RNA.

To analyze single-cell RNA sequencing data, several data quality control steps are commonly performed, such as removing both low-quality reads and doublet cells and gene read count normalization. Subsequently, clustering analysis is performed to identify cells that have similar features, referred to as cell states or cell types, often based on common expression patterns of known marker genes or novel marker genes. It should be noted that often, within one cell type, various cell states that exhibit distinct gene expression patterns can be annotated using commonly known marker genes. Probably due to differences in tissue collection, sequencing, and analysis methods, the eight studies on the human cornea showed significant variations in the total number of analyzed single cells and identified corneal cell states (Table 1) [43,44,45,46]. All studies used well-known classical marker genes to annotate different corneal cell types, and identified additional new marker genes to further define cell states within specific cell types (Table 2). We compared these studies to identify common and distinct marker genes for defining cells in the limbus, conjunctiva, central epithelium, stroma, and endothelium (Figure 2). Interestingly, many classical and some of the new marker genes to define cell states are known to be associated with corneal diseases (Table 3), demonstrating the importance of these genes in corneal biology. We will discuss these markers in detail in the following sections. However, we were unable to perform a comparison for the limbus since only the study by Collin et al. provided complete marker gene lists of limbal cells. Van Zyl et al. showed a large number of distinct marker genes compared to the other studies in every corneal region. This difference can likely be attributed to the different technology used in the van Zyl study, single-nucleus RNA sequencing, as compared to all other studies that used single-cell RNA sequencing [40]. On another note, not all of these single-cell RNA sequencing datasets are publicly available.

## 3. Classical and Novel Marker Genes Defining Cell States in the Cornea

### 3.1. The Limbus and Limbal Epithelium

A number of cell states in the limbal region of the cornea were described in the works from Català et al., Collin et al., Li et al., Ligocki et al., Maiti et al., and van Zyl et al. [28,30,31,32,33,35] (Table 2), including limbal stem cells, limbal progenitor cells, transit-amplifying cells, and limbal epithelial cells. These cell states were generally defined with marker genes that are well described for their importance in eye development and in the limbal stem cell niche, such as *KRT14*, *KRT15*, *PAX6*, *TP63*, and *MKI67* [28,30,31,32,33,35]. For instance, *KRT14* and *KRT15* are highly expressed in the limbus, particularly in limbal stem cells [49,50,51], and limbal stem cells expressing *KRT14* have been shown to play an important role in this tissue by promoting corneal wound healing [52]. *PAX6* and *TP63* are known to play important roles in the cornea, and mutations in these genes are associated with the genetic disorders Aniridia and Ectrodactyly–Ectodermal Dysplasia–Cleft Syndrome, respectively, where limbal stem cell deficiency has been reported (Table 3). Moreover, *MKI67* was used as a marker to define transit-amplifying cells in some of these studies [28,31,32,35].

In addition to these classical marker genes, a number of novel marker genes were reported in these studies. For instance, CXCL14 was proposed to be important in driving limbal cell identity both by Collin et al. and Català et al. [30,31]. Canonically, CXCL14 is hypothesized to bind to IGF1R and to be involved in AKT signaling [53,54]. This signaling pathway is important for driving the outgrowth of expanded limbal cells in vitro [55]. However, in the limbus, the location of CXCL14 was reported differently in these two studies, shown by immunostaining. Català et al. showed that CXCL14 was localized in the outer layers of the limbus, whereas Collin et al. showed that CXCL14 was localized in the limbal stem cell niche. Additionally, Collin and colleagues defined quiescent limbal stem cells (qLSC) and limbal neural crest-derived progenitor cells (LNCP) by identifying novel marker genes in these single-cell populations. LNCP expressed *CPVL* and qLSC were marked with *GPHA2* and *CDKN1B* (p27), of which the precise role in the human cornea needs to be further studied. These studies have also identified genes involved in various signaling pathways. Ligocki et al. identified pathways involved in proliferation such as the WNT [56], p53, p63 [57], and c-Myc pathways and were predicted to be important for limbal stem cells and limbal progenitor cells. Furthermore, the novel genes *CKS2*, *STMN1*, *UBE2C*, and *E2F8*, which encode transcription factors generally known to be involved in cell proliferation [58,59,60,61,62], were reported and were hypothesized to promote proliferation of TACs in the studies by Ligocki et al., Català et al., and Li et al. In the studies by Collin et al. and Gautam et al., the NF-κB pathway genes *REL*, *RELA*, and *RELB* were shown to be expressed in cell states of the limbal region [29,30]. A recent study showed that *RELA* is involved in corneal regeneration and maintenance through the retinoic acid pathway, which is implicated in age-related corneal damage [63]. Nevertheless, the exact function of the NF-κB pathway in the cornea is not fully described.

In general, transcription factors, the most important players in determining the identity of cell states [64,65], were not extensively described in these studies. Furthermore, these studies did not consistently identify the same transcription factors as the key genes to define similar cell states. For example, for defining limbal stem cells and limbal progenitor cells, Li and colleagues identified *FOXC1*, whereas this gene was not identified by Collin and colleagues. Instead, Collin and colleagues annotated *CEBPD* and *SMAD3* as key genes for the limbal stem cells and limbal progenitor cell states, which were not reported by Li and colleagues. Interestingly, all three transcription factors have been shown to be important for limbal stem cells. The role of FOXC1 and SMAD3 is to maintain the corneal epithelial identity of limbal stem cells [66,67], whereas CEBPD is known to be involved in regulating the self-renewal of these cells [68].

In some cases, classical marker genes that are commonly used for a particular cell state were identified to define different types of corneal cell states or were not identified at all. For instance, *KRT3* and *KRT12*, markers strongly associated with central corneal epithelium [69,70] were identified and used to annotate limbal epithelial cell states in the studies of Català et al., van Zyl et al., and Maiti et al. [28,31,33]. Several commonly used limbal stem cell markers, such as *ABCG2* [71,72,73], *ABCB5* [74,75,76,77], and *LRIG1* [78,79,80,81], were not reported in these single-cell RNA sequencing studies. There are several potential explanations for this. First, these limbal stem cell markers may not have been detected in these single-cell studies due to differences in mRNA and protein levels. The mRNA of these genes may decay quickly [82], but the translated protein could remain stable. This discrepancy can lead to the identification of these markers only at the protein level, as demonstrated for certain genes [83]. Second, these marker genes are often reported in cultured limbal stem cells, and the in vitro culture could affect gene expression. This is supported by a study that showed a small number of in vivo limbal stem cells expressing *ABCB5*, but when cultured and selected in vitro, this marker gene was expressed in a large number of cells [84]. Additionally, a number of studies demonstrate the importance of *ABCG2* in maintaining the in vitro culture of adult limbal stem cells [79,85,86]. Third, the expression level and importance of some marker genes could be different between model organisms. For instance, the majority of studies on *LRIG1* are performed in mice [78,80,81], and studies investigating this gene in humans are limited [79]. Nevertheless, this implies that some commonly used marker genes cannot be used to identify cell state differences in the single-cell RNA sequencing datasets.

In addition to a large variation of marker genes between studies, the cell states reported among these studies are different. For instance, no limbal stem cells or limbal progenitor cells were identified in the studies by van Zyl et al. [28]. and by Gautam et al. [29]. This could be because these studies used the entire ocular anterior segment and whole eye, respectively, for generating the data, and the number of cells specifically identified in the cornea is relatively low compared to other studies. Alternatively, single-cell analysis methods such as data filtering and clustering or the use of single-nucleus RNA sequencing could give rise to these differences [43,45]. Even in studies that specifically focused on the cornea, the identified limbal cell states still differed. For example, Collin et al. did not identify any TACs [30] that were often annotated with *MKI67* expression in other studies. However, this study reported a high expression of *MKI67* in LNCP, suggesting that these LNCP cells may be similar to TACs in other studies.

### 3.2. The Central Corneal Epithelium

As compared to the less clearly defined limbal cells, most of these single-cell RNA sequencing studies identified the central cornea epithelial cell type (Table 2). For this, a number of well-characterized membrane proteins, such as *MUC4* and *MUC16*, together with keratins [28,29,30,31,32,33,34] were used as marker genes, which are linked to corneal diseases. For example, the markers *KRT3* and *KRT12* were used in several studies to annotate stratifying central epithelium, and mutations in these two genes are associated with Meesmann corneal dystrophy (Table 3), a disorder in which the corneal epithelial layer is primarily affected, demonstrating the importance of these marker genes in this corneal layer.

In contrast to using common markers for the central epithelial cell type, studies showed consistent but also distinct marker genes to define more specific cell states in the central epithelium [28,29,30,31,32,33,34]. One consistent marker reported by Collin et al. and Català et al. to annotate basal corneal epithelium is *GJB2*, which is associated with Keratitis–Ichthyosis–Deafness Syndrome (Table 3), a disorder in which the corneal epithelium is affected, likely the basal corneal layer. Marker genes identified exclusively by Collin et al. for basal corneal epithelium include *HES1* and *HES5*, which play roles in maintaining the undifferentiated cell state [87,88]. On the other hand, *ITGB4*, *HOMER3*, and *GJB6* were uniquely chosen by Català et al. to annotate the basal epithelial cells in humans. *GJB6* has been shown to be a marker for basal corneal epithelium in rats [89]. Another example of the discrepancy in cell state annotation is wing cells. Wing cells were proposed to reside between the innermost basal layer and the outermost superficial layer of the corneal epithelium, and their function is currently unknown. In two studies, by Català et al. and by van Zyl et al., wing cells were identified using *RARRES1*, which is involved in fatty acid metabolism in epithelial cells [90], and using *CXCL17*, which is involved in maintaining homeostasis at mucosal barriers [91], respectively. However, both genes have not been previously described in the cornea, which questions whether wing cells are a biologically distinct cell type or a state within central corneal epithelial cells.

### 3.3. The Conjunctiva

Six out of the eight studies identified conjunctival cells (Table 2). Similar to the central epithelium, a number of well-characterized membrane proteins such as *MUC5AC* and *AQP5*, together with canonical conjunctiva markers *KRT13* and *KRT19* [70,92], were used as marker genes for this layer. In two studies, van Zyl et al. and Collin et al. [28,30], the common epithelial marker *KRT14* was used to annotate conjunctival cells due to its high expression in this cell type. Consistent with their findings, high expression of *KRT14* in conjunctival cells was also observed by Gautam et al. However, as *KRT14* is highly expressed in all basal epithelial cells, using *KRT14* is apparently not sufficient to define conjunctival cell type. This suggests that a combination of different markers may be necessary to define the conjunctiva cell type, or more studies need to be performed to identify conjunctiva specific markers. One potential approach could be to separate conjunctiva from the limbus and other basal cells before performing single-cell RNA sequencing, followed by confirming the locations of identified marker genes, e.g., with immunostaining.

Studies differed in the identification of novel marker genes and specific cell states in the conjunctiva. For instance, *S100A9*, a gene encoding a transmembrane protein [93], was used as a unique marker in Collin’s study to identify superficial conjunctival epithelium. Basal conjunctival epithelial cells were identified by Collin et al. and by van Zyl et al. using *S100A8* and *BCAS1*, respectively. Additionally, mucin-producing goblet cells and wing conjunctival epithelium were only identified by van Zyl and colleagues using *MUC5AC* for goblet cells, and *LCN2* and *WFDC2* for wing conjunctival epithelium. Similar to the proposed novel marker genes for other cell states, the function of these marker genes has not yet been studied in the cornea.

### 3.4. The Corneal Stroma

In general, studies made use of similar marker genes to identify the corneal stromal cell type. In the studies of Dou et al., Migocki et al., Català et al., van Zyl et al., and Collin et al., *LUM* and *KERA*, genes that are known to maintain the extracellular matrix [93,94], were used as marker genes [28,30,31,32,34]. These genes are associated with disorders that affect the stroma, i.e., posterior polymorphous corneal dystrophy, cornea plana, and congenital stromal corneal dystrophy (Table 3). Additionally, various cell states were identified within the corneal stroma, including corneal stromal stem cells, stromal keratocytes, and stromal fibroblasts [28,30,31,32,34], but the novel marker genes depicting these specific stromal cell states were largely different across the different studies. Collin et al. uniquely identified corneal stromal stem cells by using *MMP3*, *CD34*, and *ENG* (CD105). *ENG* (CD105) has been shown as a marker for vascular endothelium [95,96]. Català et al. identified several stromal keratocyte cell states, including activated, general, and transitioning keratocytes by using the keratocyte marker *ALDH3A1* [3]. For activated stromal keratocytes, defined as keratocytes playing a crucial role in maintaining the corneal stromal extracellular matrix, *COL12A1* [97] was specifically used. *MME*, together with *KERA*, were used in the study by van Zyl et al. to identify corneal stromal fibroblasts, although the role of *MME* in the cornea is still undefined. In the study by Ligocki et al., focal adhesion genes, such as *ITGB1*, *ITGB4*, *THSB1*, *THSB4*, as well as matrix metallopeptidase genes, *MMP2*, *MMP3*, and metallopeptidase inhibitor genes, *TIMP1* and *TIMP2*, were highly expressed across all stromal cell states, consistent with stromal cell function.

### 3.5. The Corneal Endothelium

The marker genes used to identify corneal endothelial cells were only partially shared between studies. In the studies by Català et al., Ligocki et al., and van Zyl et al., *ALCAM* and *CA3* were used as markers due to their high expression levels in these cells [98,99]. Other marker genes include *COL8A2* that was used in the study by Maiti et al., and *SLC4A11* used in the studies by Català et al. and Ligocki et al. *COL8A2* and *SLC4A11* seem to play significant roles in the corneal endothelium, since mutations in these genes are associated with congenital hereditary endothelial dystrophy, posterior polymorphous corneal dystrophy, and Fuchs endothelial corneal dystrophy (Table 3).

The identified cell states within the corneal endothelium differed between studies. Català and colleagues distinguished between stationary and migratory corneal endothelial cells, utilizing *COL4A3* to depict the migratory cells. In the study by van Zyl et al., pericytes and vascular endothelium were uniquely identified by expression of *NOTCH3* and *ALPL*, respectively. The role of these genes in the cornea is currently unknown, and further studies are required to investigate their functions. Intriguingly, in Collin’s study, a cell state named fibroblastic corneal endothelial cells was identified, depicted by the expression of the myofibroblast marker *ACTA2* [100,101], which was not identified in the other studies. Genes involved in signaling pathways of corneal endothelial cells were uniquely discussed in the study by Ligocki et al. [32] such as *FZD2*, involved in WNT signaling, as well as *FGF7* and *FGFR1*, which are involved in FGF signaling.

## 4. Single-Cell RNA Sequencing Applied to Human Corneal Organoids

In recent years, researchers have been studying human corneal cells through in vitro differentiation of human embryonic stem cells (hESCs) and induced pluripotent stem cells (iPSCs) into corneal lineages, either as a monolayer (2D) [84,102,103,104,105], or more recently, as multilayer (3D) systems [33,106,107,108]. Single-cell RNA sequencing is an appealing technique for 2D systems as it allows for the discovery of cellular heterogeneity and evaluation of differentiation efficiencies. However, it is particularly attractive to 3D models, such as corneal organoids, which resemble the organization and composition of the cornea, making it possible to identify the molecular profiles of the different corneal cells and to compare them to in vivo tissue. Although promising, only one study so far has published single-cell RNA sequencing analysis of human corneal organoids: Maiti and colleagues have established and analyzed three corneal organoids generated from induced pluripotent stem cells (iPSCs), cultured for 4 months and compared them to three adult human corneas [33].

In this study, the main corneal cell types were initially identified both in human corneas and corneal organoids using commonly used marker genes, e.g., *KRT5* and *KRT14* for epithelial cells, *LUM*, *DCN*, and *BGN* for stromal cells, and *TAGLN*, *TCF4*, *AQP1*, and *COL4A1/2* for endothelial cells. In general, similar cell states within these cell types were identified between human corneas and corneal organoid samples. Some *KRT14*- and *KRT15*-expressing organoid cells also expressed *TP63*, suggesting that these epithelial cells might be limbal stem cells. Likewise, conjunctival cells identified from the organoids expressed the conjunctival markers *MUC1*, *MUC4*, *MUC16*, and *MUC20* [109]. Furthermore, cells identified as corneal endothelial cells in the organoids had a similar transcriptome compared to adult corneal endothelial cells, with expression of *TAGLN*, *AQP1*, and *TCF4*.

However, there were some significant differences between organoids and in vivo tissues. First, the contribution of cell types between organoids and human corneas were different: overall, organoids presented more epithelial and endothelial cells, whereas in human corneas more stromal cells were identified. Second, the limbal progenitor marker *GPHA2* was not detected in limbal cells from corneal organoids, whereas it was expressed in cells obtained from the human cornea, consistent with the studies by Collin et al. and Ligocki et al. Third, corneal epithelial cells in organoids expressed low levels of *KRT3* and *KRT12*, whereas they were highly expressed in the human cornea. Fourth, *KERA* expression levels in stromal cells were drastically lower compared to human cornea stromal cells. Fifth, *COL8A1* and *COL8A2* were expressed in relatively low proportions in organoid endothelial cells compared to the in vivo cells. Lastly, corneal organoids seemed to have more actively dividing cells compared to in vivo corneal cells, shown by increased expression of *MKI67*, *TOP2A*, and *CCNA2*. Taken together, human corneal organoids seem to better resemble the developing cornea rather than an adult cornea, but further studies are necessary to prove the potential of these structures as representative models for human corneal development.

Nevertheless, single-cell RNA sequencing can be an excellent tool to dissect the cellular heterogeneity in organoid models and elucidate how their molecular signatures resemble in vivo human tissue.

## 5. Opportunities and Challenges of Using Single-Cell RNA Sequencing

### 5.1. Corneal Biology in Health

Single-cell RNA sequencing offers numerous opportunities to investigate the human corneal tissue as it allows to detect both common and rare cell states within a heterogeneous tissue. Additionally, single-cell RNA sequencing can detect gene expression differences between all identified corneal cell states within the tissue, and makes it possible to determine which cell states share similarities and which are more distinct from each other. Traditional bulk methods mask differences between these cell types. Moreover, detecting gene expression differences between different corneal cell states can be crucial for understanding cell function and regulation within the cornea. From a developmental point of view, single-cell RNA sequencing can provide fundamental insights into multi-cell fate acquisition, gene regulatory networks or how different cell states are transitioning during development [110,111,112].

In addition to understanding biology of the in vivo tissues, single-cell RNA sequencing can be used to improve in vitro models, such as differentiation studies of ex vivo tissue [113,114,115] or from iPSCs [84,116,117]. The transcriptomic profiles of heterogeneous cell populations can be compared to in vivo cells and used to further optimize differentiation strategies. Likewise, this approach can be applied in generating more complex in vitro corneal models, both in 2D, such as limbal stem cells [84,118], or 3D, e.g., corneal organoids (described above).

Furthermore, single-cell RNA sequencing can be combined with other single-cell omics approaches to uncover molecular mechanisms of cells in cornea tissues. Firstly, combining single-cell RNA sequencing and single-cell Assay for Transposase-Accessible Chromatin (ATAC) sequencing, a method for detecting regulatory sequences in the genome [119], provides valuable insights into the mechanisms that regulate gene expression. This approach makes it possible to identify the key transcription factors that define different states within corneal cells. Secondly, single-cell RNA sequencing could be combined with single-nucleus RNA sequencing. Because these two methods measure levels of different RNA species, processed cytoplasmic RNA and unprocessed nuclear RNA, comparing the results from these methods will provide valuable insights into RNA processing. Finally, single-cell RNA sequencing data can be combined with spatial transcriptomics [120] to determine the exact location of identified markers and cell states [121,122] in human cornea tissues. Altogether, future studies empowered by single-cell RNA sequencing will allow us to better understand corneal biology.

Single-cell RNA sequencing of the cornea is still accompanied by a number of challenges. The first challenge is that detecting cell states within a tissue depends on many factors such as sample collection, dissection, and dissociation (Table 2) as well as downstream analysis pipelines. For example, using the whole eye for single-cell analyses seems to make it difficult to identify a large number of corneal cell states (Table 1). One potential option to simplify the analysis is to dissect the cornea into smaller tissue layers, as demonstrated by Català and colleagues for the epithelium, stroma, and endothelium [31]. By doing so, cells can be assigned to their respective dissected tissue layers. Besides, data pre-processing, batch correction, and clustering methods [43,44,123] may influence cell state detection. Overall, it remains challenging to identify and distinguish differences between specific cell states in all cell types (Table 2), as the current knowledge of specific marker genes is still insufficient. It is expected that cell state differences may be distinguished by the use of transcription factors, the main drivers of cell fate determination, but unfortunately, this is not yet straightforward. A prime example of this is *PAX6*. *PAX6* is essential for eye development and corneal homeostasis [124], but due to this gene being highly expressed within the limbus [125], it would not be identified as a key gene for individual limbal cell states and therefore cannot be used to annotate different cell states in this tissue. Next, it is still challenging to determine whether certain cell states are biologically distinct or whether they are a result of technical differences such as tissue dissection or data analysis [43,44,123], even when marker genes have been identified. This is also because the function of some proposed marker genes remains unknown, which further emphasizes the importance of studying the biological and functional relevance of the newly proposed corneal cell states and marker genes in follow-up studies of single-cell RNA sequencing.

### 5.2. Corneal Biology in Disease

Single-cell RNA sequencing has high potential in investigating corneal diseases and exploring opportunities for treatment options as it allows the analysis of complete corneas or large tissue samples. Traditional approaches mainly focused on affected cells or tissues, e.g., using the corneal endothelium to study Fuchs endothelial corneal dystrophy [126,127,128]. Although direct investigation of the affected tissue often generates valuable insights into disease mechanisms, important information regarding molecular, cellular, or physiological changes affecting the nearby environment may be lost. For example, the stroma has been shown to be important for maintaining corneal epithelial cell function [129,130]. Therefore, when studying stroma-related conditions, e.g., keratoconus [34,131,132], epithelial defects may be overlooked if only the stroma is investigated.

As cell states could be rare and difficult to be physically separated, single-cell RNA sequencing enables the determination of which cell states within the tissue are present or absent in corneal diseases [34,127], as compared to healthy conditions. Additionally, comparisons of specific cell states between healthy and disease conditions can help to identify genes driving disease phenotypes specific to cell states [34,38]. This cell state specific approach could also allow exploration of treatment options targeting specific cell states associated with corneal disease phenotype [133]. This is impossible to achieve with bulk RNA sequencing methods.

Identifying corneal cell states in disease with single-cell RNA sequencing poses additional challenges beyond those described for healthy cells. Marker genes defining healthy cell states may be differentially expressed in corneal diseases, making it difficult to use them to identify cell states and to perform downstream analyses. Furthermore, the standard method for differential gene expression analysis in single cell RNA sequencing, called Wilcoxon ranked-sum test [134,135], is probably not optimal to detect differential gene expression between healthy and diseased corneal cells. This is because this method is best suited for determining differences between two independent sample groups, e.g., cells from different cell states, and may introduce false positives when directly comparing similar cell states [136]. Therefore, alternative methods should be considered, such as edgeR [137], DESeq2 [138], and limma [139], which are able to perform differential gene expression based on aggregated single-cell data.

## 6. Concluding Remarks

Single-cell RNA sequencing is a powerful tool to identify genes that drive cell identity at a single-cell resolution. Using this technology, several studies have so far provided resources of marker genes and cell states of the human cornea. In addition to many canonical marker genes important for various corneal cell types that were confirmed, new marker genes were described. These studies also reported both common and rare cell states within different cell types and determined gene expression differences between them. However, probably due to technical aspects of each analysis, such as different sample collection and dissociation or analysis methods, the identified marker genes and cell states varied greatly among these studies. These inconsistencies make the use of these resources difficult for downstream application in the research field of corneal biology and disease. One solution to the problem is to construct a meta-atlas of the cornea by integrating all currently publicly available single-cell RNA sequencing data of the human cornea and to provide a comprehensive marker gene and cell state reference.

It is expected that single-cell RNA sequencing will continue to advance corneal research. In combination with other methods, this technology will further elucidate regulatory mechanisms of corneal cells. This technology can also help to optimize in vitro models and cell differentiation protocols via a data-driven approach. Finally, by comparing cell states and gene expression patterns in health and disease at the single-cell level, it is likely to identify genes and pathways affected in specific cell states in corneal diseases and to assist the development of targeted strategies for treatment [140].

## Figures and Tables

**Figure 1 cells-12-01808-f001:**
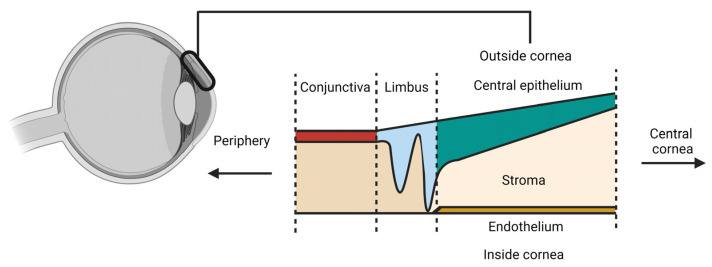
Schematic overview of the corneal tissue layers. The cellular layers are highlighted, with the limbus acting as the border between the conjunctiva and the central cornea.

**Figure 2 cells-12-01808-f002:**
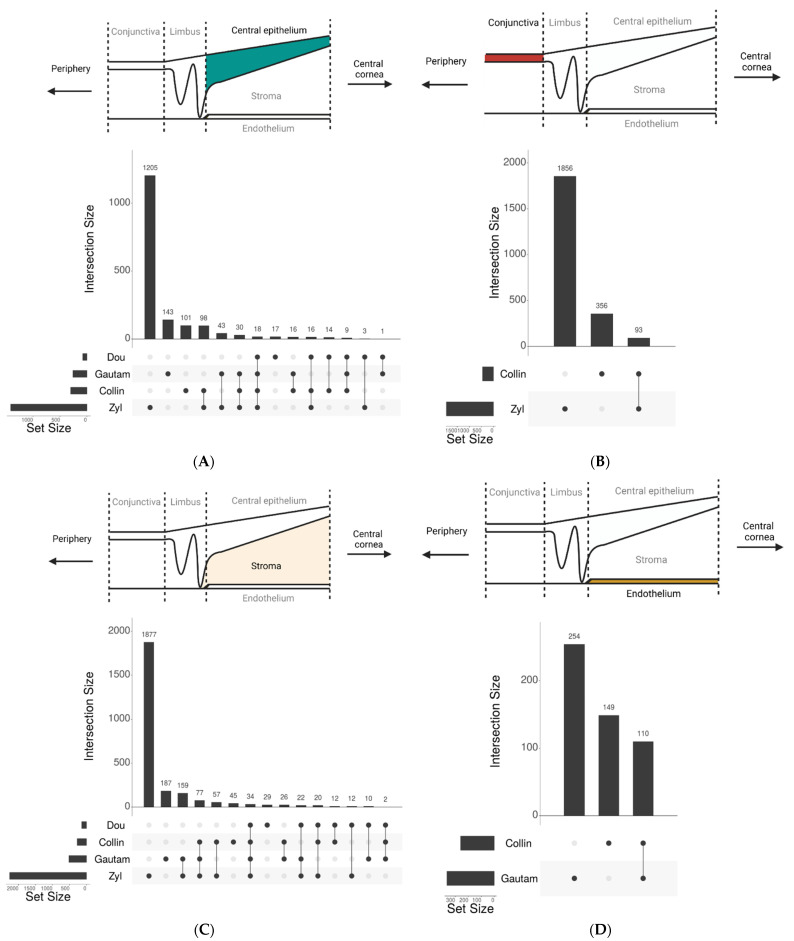
Upset plots [47] of intersecting marker genes between studies. Marker genes for cells in (**A**) Central epithelium; (**B**) Conjunctiva; (**C**) Corneal stroma; (**D**) Endothelium are summarized from studies that reported marker genes for these regions. The gene lists were obtained from supplementary data tables from corresponding studies. The vertical intersection size represents the number of identified marker genes to define cell states in each study or the intersection with other studies specified below the graphs. The connected dots indicate which studies share intersection of marker genes. The horizontal set size indicates the total number of marker genes identified in each study. Note that only Collin et al. provided the complete list of limbal marker genes in the format of supplementary table, and therefore, no comparison between studies could be performed for the limbus. The number of marker genes was counted based on selected genes with adjusted *p*-value smaller than 0.05 and a log2 (fold change) greater than zero [28,29,30,34].

**Table 1 cells-12-01808-t001:** Summary of single-cell RNA sequencing studies on the human cornea. The asterisk indicates the study using single-nucleus RNA sequencing.

Study and Year of Publication	Tissues	Total Number of Cells	Total Number of Corneal Cell States	Raw Data Publicly Available	Number of Donors	Dissociation Method
Zyl (2022) * [28]	Ocular anterior segment	191,992	28	Yes	6	Separate dissection
Gautam (2021) [29]	Complete eye	Approx. 50,000	7	Yes	3	Separate dissection
Collin (2021) [30]	Complete cornea	213,430	21	Yes	6	Bulk enzymatic
Català (2021) [31]	Complete cornea	19,472	15	Yes	8	Separate dissection
Ligocki (2021) [32]	Complete cornea	16,234	16	No	6	Bulk enzymatic
Maiti (2022) [33]	Central and peripheral cornea	53,438	12	Yes	3	Separate dissection
Dou (2022) [34]	Central cornea	39,214	6	No	4	Bulk enzymatic
Li (2021) [35]	Limbus	16,360	12	Yes	2	Bulk enzymatic

**Table 2 cells-12-01808-t002:** Summary of identified corneal cell states of the human cornea using single-cell RNA sequencing studies. Abbreviations: LSC—Limbal stem cells; qLSC—Quiescent LSC; LNCPs—Limbal neural-crest derived progenitor cells; LPCs—Limbal progenitor cells; CSSCs—Corneal stromal stem cells; FCECs—Fibroblastic corneal endothelial cells. The asterisk indicates the study using single-nucleus RNA sequencing.

Layer	Study	Names of Corneal Cell States
Limbus	Català (2021) [31]	Cells in the limbal stem cell niche, corneal basal limbal epithelial cells, terminally differentiated migratory limbal epithelial cells, transiently amplifying cells, LSC in the limbal stem cell niche or in the peripheral cornea, wing superficial limbal epithelial cells
	Collin (2021) [30]	qLSC, LNCPs, LPCs, limbal fibroblasts, limbal stroma cells, limbal stromal keratocytes, limbal superficial epithelium, limbal suprabasal cells
	Li (2021) [35]	Progenitor cells, transiently amplifying cells
	Ligocki (2021) [32]	Early limbal progenitor cells, late limbal progenitor cells, transiently amplifying cells
	Maiti (2022) [33]	Limbal progenitor cells, limbal epithelial cells, corneal epithelial stem cells
	Zyl (2022) * [28]	Basal limbal epithelial cells, wing limbal epithelial cells, superficial limbal epithelial cells, transiently amplifying cells
Corneal epithelium	Català (2021) [31]	Basal corneal epithelium, terminally differentiated central corneal epithelium, wing superficial epithelial cells
	Collin (2021) [30]	Basal corneal epithelium, central cornea suprabasal cells, suprabasal corneal epithelium
	Dou (2022) [34]	Corneal epithelial cells
	Gautam (2021) [29]	ELF3-high corneal epithelial cells, TGFBI-high corneal epithelial cells
	Li (2021) [35]	Differentiated cells
	Ligocki (2021) [32]	Basal epithelial cells, central superficial mature epithelial cells, transitional epithelial cells, transiently amplifying cells
	Maiti (2022) [33]	Corneal epithelial stem cells, differentiated corneal epithelium, differentiated superficial corneal epithelium
	Zyl (2022) * [28]	Basal corneal epithelium, superficial-most squamous epithelium, wing superficial cells
Conjunctiva	Collin (2021) [30]	Basal conjunctival epithelium, superficial conjunctival epithelium
	Gautam (2021) [29]	Conjunctival cells
	Li (2021) [35]	Conjunctiva
	Ligocki (2021) [32]	Conjunctival epithelial cells
	Maiti (2022) [33]	Conjunctival epithelial cells
	Zyl (2022) * [28]	Basal conjunctival epithelium, mucin-producing goblet cells, superficial conjunctival epithelium, wing conjunctival epithelium, conjunctival melanocytes
Stroma	Català (2021) [31]	Activated stromal keratocytes, general stromal keratocytes, transitioning keratocytes in the stroma to myofibroblasts
	Collin (2021) [30]	CSSCs, central stromal keratocytes
	Dou (2022) [34]	Corneal stroma cells
	Ligocki (2021) [32]	Stromal cells
	Maiti (2022) [33]	Corneal stromal cells
	Zyl (2022) * [28]	Corneal stromal fibroblasts, corneal stromal keratocytes
Endothelium	Català (2021) [31]	Corneal endothelium stationary cells, corneal endothelium migratory cells
	Collin (2021) [30]	Corneal endothelium, FCECs
	Ligocki (2021) [32]	Corneal endothelial cells
	Maiti (2022) [33]	Corneal endothelium
	Zyl (2022) * [28]	Endothelial lining, pericytes, vascular endothelium

**Table 3 cells-12-01808-t003:** Marker genes and their cell states reported in single-cell studies that are associated with cornea-related diseases. OMIM numbers were retrieved from the Online Catalog of Human Genes and Genetic Disorders [48].

Markers Identified in Single-Cell Studies	Cell States and Studies	Associated Disorder	Affected Cornea Layer	Gene OMIM Number
*GJB2*	Wing superficial limbal epithelial cells (Català), basal corneal epithelium (Català), basal corneal epithelium (Collin)	Keratitis–Ichthyosis–Deafness (KID) Syndrome	Epithelium, stroma	121011
*TP63*	Limbal progenitor cells (Collin), LNCP (Collin), limbal suprabasal cells (Collin), limbal epithelial stem cells (Català), epithelial stem cells (Maiti)	Ectrodactyly–Ectodermal Dysplasia–Cleft Syndrome	Epithelium	603273
*PAX6*	LNCP (Collin), corneal epithelium (Zyl)	Aniridia	Epithelium	607108
		Axenfeld–Rieger Syndrome	Neural-crest derived structures, stroma, endothelium	601542
*PITX2*	Corneal endothelium stationary cells (Català), corneal endothelium migratory cells (Català)			601090
*FOXC1*	Limbal stem cells (Li)			601542
*TGFBI*	TGFBI-hi corneal epithelial cells (Gautam)	Epithelial basement membrane dystrophy	Epithelium	601692
*KRT12*	Basal limbal epithelial cells (Zyl), limbal epithelial basal cells (Català), terminally differentiated migratory limbal epithelial cells (Català), basal and wing cells (Zyl),central superficial mature epithelial cells (Ligocki), corneal epithelium (Dou and Zyl), differentiated cells (Li), differentiated corneal epithelium (Maiti), transitional epithelial cells (Ligocki), wing superficial central epithelium (Català)	Meesmann corneal dystrophy		601687
*KRT3*				148043
*DCN*	Corneal stromal cell subsets (Maiti), stromal cells (Ligocki and Dou)	Congenital stromal corneal dystrophy	Stroma	125255
		Posterior amorphous corneal dystrophy		
*LUM*	Activated stromal keratocytes (Català), corneal stromal fibroblasts (Zyl), CSSCs (Collin), central stromal keratocytes (Collin), general stromal keratocytes (Català), stromal cells (Ligocki and Dou), transitioning keratocytes stromal myofibroblasts (Català), corneal stromal cell subsets (Maiti), corneal endothelium (Maiti)			600616
*KERA*		Cornea plana	Stroma, Descemet membrane	603288
*COL8A2*	Corneal endothelium (Maiti)	Fuchs endothelial and posterior polymorphous corneal dystrophy	Endothelium, Descemet membrane	120252
*SLC4A11*	Corneal endothelium stationary and migratory cells (Català), corneal endothelial cells (Ligocki)	Congenital hereditary and Fuchs endothelial corneal dystrophy	Endothelium	610206

## Data Availability

The data presented in this study containing single-cell RNA sequencing data of the human cornea are publicly available, described in the publications of these single-cell studies [28,29,30,31,32,33,34,35].

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
