# Peer review of "Single-Cell RNA Sequencing: Opportunities and Challenges for Studies on Corneal Biology in Health and Disease"

_cells, 2023, doi:10.3390/cells12131808_

Round 1

Reviewer 1 Report

The manuscript "Single-Cell RNA-Sequencing: Opportunities and Challenges 2 for Studies on Corneal Biology in Health and Disease" by Arts et al. review works on single cell RNA sequencing in the cornea. I think this paper is very important to introduce corneal researches to this powerful technique. As the authors discuss; single cell RNA sequencing can detect different cell states within the same cell types, that it is crucial for the understanding of complex diseases such as senescence, aging related diseases, etc. 

I suggest minor revisions:

Table1. The asterisk is missing.

Table 2. The asterisk is missing.

Table 3. SLC4A11 is associated as well with Fuchs endothelial corneal dystrophy.

5.1. Corneal biology in health.  Can you provide an example and a reference where the 3 layers (epithelium, stroma and endothelium) were first isolated by dissection or cryosection, then single cell suspension generated and analyzed by single cell RNA sequencing? This may be another approach to simplify the analysis. 

Reviewer 2 Report

The manuscript by Julian Artsand co-workers summarizes for the first time studies of human cornea using single cell transcriptomic profiling (scRNA-seq) coupled with computational biology analyses and modeling. To increase broader impact of the manuscript, it is adviced to extend section 2 by a new introductory paragraph on history of scRNA-seq and pioneering studies of retinal ganglion cells and the global impact of these eye studies for this emerging new field. Taken together, this manuscript addresses highly powerful but still imperfect experimental platform to understand complexity of the human cornea and to close some remaining and important gaps in our knowledge of corneal biology & physiology. This manuscript is very appropriate for publication in the special issue of Cells “Stem Cell Biotechnology in Ocular Regenerative Medicine and Drug Discovery” following some minor editing and inclusion of additional general references. 

Additional comments:

1)    Introduction: Include the role of cornea in light refraction via the cornea-lens “refractone” (e.g. PMID: 31575608).

2)    Introduction: Provide a separate paragraph on corneal development (e.g. PMID: 26310148, 25130543, 33039457 and 32697979) and major signaling pathways involved.

3)    Introduction: Define embryonic origin of each corneal epithelium, stroma, endothelium, and conjunctiva.

4)    Table 2: In column “Study”, include year of the publication.

5)    Reformate the final manuscript to prevent splitting Table 3 into two pages.

6)    Section 4: Add relevant studies such as SEAM procedure published in Nature 2016 (PMID: 26958835) and others (e.g. Kamarudin et al. 2018, Mikhailova et al. 2014; Lee et al. 2021, …).

7)    There is a paper “in press” by Chakravarti lab in PNAS Nexus on human corneal organoids.

8)    Multiple places: Provide examples where mRNA levels do not correlate with protein levels and that some mRNAs may be transitionally retained in the nuclei prior their export to the cytoplasm and translation.

9)    Include list of abbreviations.
